# The Impact of Prior Antithrombotic Use on Blood Viscosity in Cardioembolic Stroke with Non-Valvular Atrial Fibrillation

**DOI:** 10.3390/jcm12030887

**Published:** 2023-01-22

**Authors:** Yo-Han Jung, Sang-Won Han, Joong-Hyun Park

**Affiliations:** 1Department of Neurology, Gangnam Severance Hospital, Yonsei University College of Medicine, Seoul 03722, Republic of Korea; 2Department of Neurology, Inje University College of Medicine, Seoul 04551, Republic of Korea

**Keywords:** anticoagulants, atrial fibrillation, stroke, viscosity

## Abstract

Although clinical studies have demonstrated that prior use of antiplatelets was associated with decreased blood viscosity (BV) in patients with acute ischemic stroke, the impact of previous anticoagulant use on blood viscosity in cardioembolic stroke with non-valvular AF (NVAF) has not yet been clearly studied. This single-center retrospective observational study aimed to determine the impact of prior antithrombotic (antiplatelet and anticoagulant) use on BV in patients with cardioembolic stroke (CES) due to NVAF. Patients with CES and NVAF were analyzed with the following inclusion criteria: (1) patients over 20 years of age admitted within five days of stroke onset; (2) ischemic stroke presumably due to an NVAF-derived embolus; (3) compatible cortical/subcortical lesion on brain computed tomography or magnetic resonance imaging; (4) hemoglobin level of 10–18 mg/dL; and (5) receiving antiplatelets within five days or anticoagulants within two days if previously medicated. From the screening of 195 patients (22% of the total stroke population during the study period) who had experienced ischemic stroke with AF, 160 were included for the final analysis. Eighty-nine patients (56%) were taking antithrombotics (antiplatelet, 57%; warfarin, 13%; NOACs, 30%) regularly. Compared to patients without previous antithrombotic use, those with previous antithrombotic use (antiplatelets, warfarin, and NOACs) were significantly associated with decreased systolic BV (SBV) and diastolic BV (DBV) (*p* < 0.036). In multiple linear regression analysis, hematocrit (Hct) level and prior antithrombotic use were significantly associated with decreased SBV and DBV. Hct was positively correlated with increased SBV and DBV. In Hct-adjusted partial correlation analysis, prior uses of any antithrombotic agents were associated with decreased SBV (r < −0.270, *p* < 0.015) and DBV (r < −0.183, *p* < 0.044). In conclusion, this study showed that prior antithrombotic use (antiplatelets, VKAs, and NOACs) was associated with decreased SBV and DBV in patients presenting with acute CES secondary to NVAF. Our results indicated that previous use of NOACs may be a useful hemorheological parameter in patients with acute CES due to NVAF. Accumulation of clinical data from a large number of patients with the risk of stroke occurrence, initial stroke severity, and functional outcome is necessary to assess the usefulness of BV.

## 1. Introduction

Cardioembolic stroke (CES), which accounts for one-fifth of all ischemic strokes, generally has a poor outcome [1,2]. Among potential cardiac sources of embolism, non-valvular atrial fibrillation (NVAF) represents the most common sustained cardiac arrhythmia and the main cause of CES [2]. In current practice, oral anticoagulants (OACs), including vitamin K antagonists (VKAs) and new oral anticoagulants (NOACs), are effective medications for primary and secondary stroke prevention in patients with atrial fibrillation (AF) [3,4]. Similar to VKAs, the efficacy and safety of NOACs in ischemic stroke prevention are well established. Recent studies suggested that prior antithrombotic use was associated with less severe neurological deficits and favorable functional outcomes for stroke in AF patients [5,6].

Blood viscosity (BV), the key predictor of endothelial shear stress, is defined as intrinsic resistance applied to blood flow [7]. BV determines the frictional force applied to the blood vessel wall and is characterized by blood stickiness and thickness [7,8]. As an important hemorheological parameter, BV is determined by hematocrit (Hct) level, plasma viscosity (PV), and erythrocyte deformability [7]. Diastolic BV (DBV), which represents the hemorheological state of blood corresponding to a very slow flow, becomes a critical factor for determining the tissue perfusion status and could more considerably impact microvascular perfusion than systolic BV (SBV) in acute ischemic stroke [9]. Elevated BV, which involves increased Hct and PV, and decreased erythrocyte deformability, increases thromboembolic risk. Therefore, elevated BV is related to an increased risk of cardiovascular events and may contribute to ischemic stroke occurrence, in addition to all-cause mortality [9,10].

In AF patients, several studies suggested that hyperviscosity due to hemorheological alterations was frequent [9,11,12]. One study showed that increased BV at high shear rates and reduced erythrocyte deformability were independently related to ischemic stroke occurrence in AF patients [9]. Regarding treatment, VKAs reduce BV significantly in patients with acute ischemic stroke and NVAF [11]. We previously demonstrated that prior use of antiplatelets was associated with decreased BV in patients with acute ischemic stroke [13]. However, the impact of prior antithrombotic use on BV in patients with NVAF has not yet been elucidated. Therefore, this study aimed to determine the effects of prior antithrombotic use on BV in patients with acute CES due to NVAF.

## 2. Materials and Methods

### 2.1. Patients

A single-center retrospective observational study was conducted on CES patients with NVAF from March 2020 to February 2022. The inclusion criteria were (1) patients over 20 years of age admitted within five days of stroke onset; (2) ischemic stroke presumably due to an embolus arising from NVAF according to the Trial of ORG 10172 in the Acute Stroke Treatment Classification [14]; (3) compatible cortical/subcortical lesion on brain computed tomography (CT) or magnetic resonance imaging (MRI); (4) hemoglobin (Hb) level between 10–18 mg/dL; and (5) taking antiplatelets within five days or anticoagulants within two days if patients were on medication previously. Patients were excluded if they had (1) intravenous (IV) thrombolysis or intra-arterial thrombectomy; (2) moderate to severe cardiac valvular disease, prosthetic valve, or congenital heart disease; (3) a history of cancer or any thromboembolic events other than stroke; (4) concomitant use of antiplatelets and anticoagulants; and (5) two or more potential causes of stroke according to the Trial of ORG 10172 in the Acute Stroke Treatment Classification [14].

All patients underwent systemic investigations at admission. Traditional vascular risk factors, medical history, and laboratory findings were assessed. Complete blood counts, renal and liver function, blood lipid profiles, coagulation factors, and electrocardiography were examined by standard methods. Brain CT or MRI were performed for every patient. During admission, transthoracic echocardiography was performed, and only patients with NVAF were enrolled. Medications taken regularly during the week preceding admission were recorded by a skilled pharmacist on the basis of the hospital records. This study was approved by the Inje University Sanggye Paik Hospital Research Ethics Committee (IRB No. 2019-04-018-002). The informed consent requirement was waived because the database was accessed only for analysis. This study removed individual identifying variables, and the researcher could secure only the results.

### 2.2. BV Measurement

BV was measured by previously described methods [13,15]. Briefly, whole BV (WBV) was measured using a computerized scanning capillary-tube viscometer (SCTV) (Hemovister, Pharmode Inc., Seoul, Republic of Korea). SCTV measured WBV over a wide range of shear rates, ranging from 1 to 1000 s^−1^. In this study, BV measured at shear rates of 300 s^−1^ and 1 s^−1^ were selected as the SBV and DBV, respectively. SBV and DBV were characterized by viscosities at high and low shear rates, respectively [16]. Laboratory tests, such as BV, Hb, and Hct, were collected before IV hydration therapy.

### 2.3. Statistical Analysis

Baseline demographic and laboratory characteristics were reported as numbers (percentages) or means and standard deviations. The normality distribution of variables was estimated using the Kolmogorov–Smirnov test. Differences in baseline characteristics between patients without prior antithrombotic use and those with prior antithrombotic use were compared using an independent sample t-test or the Mann–Whitney U test for quantitative variables and the chi-squared test for categorical variables. The baseline parameters among patients without prior antithrombotic use, those with prior antiplatelet use, those with prior warfarin use, and those with previous NOACs use were analyzed using a one-way analysis of variance with a Tukey post hoc test for continuous variables, as applicable. Univariate linear regression models between the dependent variable (BV) and the baseline characteristics were investigated. Multivariate linear regression models were used to investigate the independent relationship between BV and significant variables in the univariate analysis. Partial correlation analysis was used to evaluate the correlation between BV and each previously used antithrombotic agent after removing the effects of Hct. Statistical significance was defined as a two-sided *p*-value < 0.05. Data analysis was performed using SPSS version 25.0 (IBM Co., Armonk, NY, USA) for Windows.

## 3. Results

A total of 195 patients (22% of the total ischemic stroke patients during the study period) who had experienced ischemic stroke with AF were screened for study enrolment, and 160 were included in the final analysis. Table 1 shows the demographic, clinical, and laboratory characteristics of the enrolled patients. The mean age was 77.3 ± 10.31 years, and 49% of the patients were women. Eighty-nine patients (56%) regularly took antiplatelets within five days or OACs within two days of symptom onset (antiplatelets, 57%; warfarin, 13%; NOACs, 30%). Compared to the patients without antithrombotic use, patients with prior antithrombotic use were older. They also had higher INR, hypertension, stroke, coronary artery disease (CAD), and statin use rates, as well as lower Hb, Hct, total cholesterol (TC), low-density lipoprotein cholesterol (LDL-C), SBV, and DBV levels, which were likely related to their use of antithrombotics and statins. The baseline characteristics of men in the two groups were similar to those of the total population. Unlike the total population, no significant differences in age; hypertension; CAD; statin use rates; and Hb, Hct, TC, and LDL-C levels were observed in women between the two groups. SBV and DBV significantly decreased with prior antithrombotic use in both men and women.

The differences in baseline characteristics between patients without antithrombotic use and each group with prior use of a particular antithrombotic agent is shown in Table A1. Compared to the group without prior antithrombotic use, patients with previous use of antiplatelets were older. They also had hypertension, stroke, CAD, and statin use rates, as well as lower Hb, Hct, TC, LDL-C, SBV, and DBV levels. Higher INR and lower hs-CRP, SBV, and DBV were observed in patients taking warfarin. In the NOAC group, patients had higher INR, stroke, and statin use rates, and lower Hb, Hct, SBV, and DBV levels, which were likely related to the use of NOACs. SBV and DBV were all significantly decreased in the groups with previous uses of antiplatelets, warfarin, and NOACs as compared to the group without prior antithrombotic use. Table A2 shows the baseline characteristics of the study population according to the type of antithrombotic agents. The group with previous use of antiplatelets was older and had CAD in comparison with the groups with previous uses of warfarin and NOACs. INR was higher in the groups with previous uses of warfarin and NOACs than in the group with previous use of antiplatelets. There were no differences in SBV and DBV among antiplatelet, VKA, and NOAC groups.

Multiple linear regression analysis revealed that Hct level and prior antithrombotic use were significantly associated with decreased SBV and DBV. Hct was positively correlated with increased SBV and DBV. Compared to patients without prior antithrombotic use, all groups with previous uses of antithrombotic agents were significantly associated with decreased SBV and DBV (Table 2). In Hct-adjusted partial correlation analysis, previous uses of all antithrombotic agents were associated with decreased SBV and DBV (r < −0.270, *p* < 0.015, and r < −0.183, *p* < 0.044, respectively).

## 4. Discussion

This study investigated the impact of prior antithrombotic use on BV in patients presenting with acute CES secondary to NVAF. Interestingly, compared to the group without prior antithrombotic use, all groups with prior antithrombotic use (antiplatelets, VKAs, and NOACs) were significantly associated with decreased SBV and DBV. There was no difference in SBV and DBV according to the type of antithrombotic agents used.

Approximately 20% of ischemic strokes are associated with cardioembolism, and the majority of CES cases are precipitated by AF. AF increases stroke risk by approximately five times [2]. To reduce stroke risk, current guidelines recommend the use of OACs with a preference for NOACs in most patients with NVAF. While warfarin reduces stroke by 60–70%, NOACs also provide a similar stroke reduction rate with the appropriate dose and medication adherence [5]. Anticoagulants inhibit the activities of coagulation factors and prevent blood clotting. Unlike traditional VKAs, NOACs directly inhibit key proteases (factors IIa and Xa). The advantages of NOACs are their high efficacy in reducing intracranial hemorrhage, similar rates of major bleeding, the convenience of use, and the absence of laboratory monitoring requirements [2].

Endothelial damage, impaired bloodstream, and hypercoagulability trigger thrombus formation [17]. Considering the Virchow triad, thromboembolic susceptibility in AF may be associated with structural heart disease, blood stasis in the atria, and hyperviscosity due to hemorheological alterations. AF causes a structural change in the left atrium (LA) and LA appendage (LAA), which increases the likelihood of an impaired bloodstream. AF also changes the composition of clotting factors, such as fibrin and thrombin-antithrombin complexes, which promote thrombus formation [11]. BV represents a primary mechanism for thrombus formation [18]. Increased BV, the key predictor of endothelial shear stress, could trigger thrombus formation [19]. A previous study showed that increased SBV and reduced erythrocyte deformability were independently associated with cerebral infarction in AF patients, suggesting that erythrocyte alterations themselves might be involved in the pathophysiology of ischemic events in AF [9]. Increased BV is also associated with the severity of spontaneous echo contrast in stroke patients and elevates the risk of LA and LAA thrombus [20,21]. Warfarin, heparin, and argatroban can decrease BV [22]. While one study showed that warfarin, but not aspirin, reduced BV significantly in patients with acute ischemic stroke with NVAF [11], the relationship between NOACs and BV in AF has not yet been reported. Our study clearly demonstrated that prior NOACs use was associated with decreased BV in patients with acute CES secondary to NVAF. Prior NOACs use may be a useful hemorheological parameter in AF.

Regarding antiplatelet agents, different results have been reported, depending on the study design and the enrolled population [11,13]. In this study, previous uses of aspirin and aspirin with clopidogrel reduced BV. These results were consistent with those from previous studies, which showed that while aspirin and cilostazol did not change BV, lower BV responses were observed with dipyridamole and clopidogrel treatment [11]. We believe that an exhaustive comparison of results from the previous reports was difficult due to differences in the study designs. Variance in the patient population and underlying disease may contribute to discrepancies in the findings and should be considered when interpreting the study results.

Our study had several limitations. First, the number of patients included the prior NOAC group (edoxaban n = 17, rivaroxaban n = 4, apixaban n = 3, and dabigatran n = 3) was not large in this retrospective study, which could imply the existence of unmeasured biases and affect the validity of the results. We also were unable to obtain important clinical data, such as initial stroke severity and outcome. Therefore, we cannot demonstrate whether those SBV and DBV in patients with CES differences have clinical implications. Second, the dose-dependent manner of BV reduction according to the activity of VKAs and NOACs was unable to be evaluated due to the lack of specific coagulation tests. Third, the statistical power to support our conclusion was fairly weak, mostly due to the small sample size and the lack of age-matched controls. Finally, we enrolled only Korean patients, which limited our ability to generalize the findings to global populations. These limitations should be considered when interpreting our study results.

## 5. Conclusions

This study showed that prior antithrombotic use (antiplatelets, VKAs, and NOACs) was associated with decreased SBV and DBV in patients presenting with acute CES secondary to NVAF. Our results indicated that previous use of NOACs may be a useful hemorheological parameter in patients with acute CES due to NVAF. Accumulation of clinical data from a large number of patients with the risk of stroke occurrence, initial stroke severity, and functional outcome is necessary to assess the usefulness of BV.

## Figures and Tables

**Table 1 jcm-12-00887-t001:** The baseline characteristics of the enrolled population.

	Total (n = 160)				Men (n = 81)				Women (n = 79)			
Prior Antithrombotics	Total	No (n = 71)	Yes (n = 89)	*p*	Total	No (n = 40)	Yes (n = 41)	*p*	Total	No (n = 31)	Yes (n = 48)	*p*
Age, years	77.3 ± 10.31	74.9 ± 10.71	79.1 ± 9.63	0.011 *	74.6 ± 10.2	72.0 ± 9.60	77.2 ± 10.32	0.020 *	79.9 ± 9.71	78.8 ± 11.00	80.7 ± 8.80	0.390
Female	79 (49)	31 (43.7)	48 (53.9)	0.197	0 (0)				100 (100)			
Hypertension	123 (76.9)	48 (67.6)	75 (84.3)	0.013 *	61 (75.3)	24 (60.0)	37 (90.2)	0.002 *	62 (78.5)	24 (77.4)	38 (79.2)	0.854
Diabetes mellitus	45 (28.1)	18 (25.4)	27 (30.3)	0.486	19 (23.5)	8 (20.0)	11 (26.8)	0.468	26 (32.9)	10 (32.3)	16 (33.3)	0.921
Dyslipidemia	103 (64.4)	41 (57.7)	62 (69.7)	0.118	46 (56.8)	20 (50.0)	26 (63.4)	0.223	57 (72.2)	21 (67.7)	36 (75.0)	0.482
Stroke	43 (26.9)	10 (14.1)	33 (37.1)	0.001 *	26 (32.1)	8 (20.0)	18 (43.9)	0.021 *	17 (21.5)	5 (6.5)	15 (31.3)	0.009 *
Coronary artery disease	17 (10.6)	3 (4.2)	14 (15.7)	0.021 *	12 (14.8)	2 (5.0)	10 (24.4)	0.028 *	5 (6.3)	1 (3.2)	4 (8.3)	0.643
Current smoking	17 (10.6)	9 (12.7)	8 (9.0)	0.452	16 (19.8)	8 (20.0)	8 (19.5)	0.956	1 (1.3)	1 (3.2)	0 (0)	0.392
Statins use	93 (58.1)	34 (47.9)	59 (66.3)	0.019 *	43 (53.1)	17 (42.5)	26 (63.4)	0.059	50 (63.3)	17 (54.8)	33 (68.8)	0.210
Hemoglobin, g/dL	13.4 ± 1.88	13.99 ± 1.92	12.85 ± 1.70	<0.0001 *	14.0 ± 1.91	14.7 ± 1.86	13.4 ± 1.74	0.002 *	12.7 ± 1.61	13.1 ± 1.63	12.4 ± 1.55	0.065
Hematocrit, %	40.5 ± 5.39	42.21 ± 5.51	39.18 ± 4.92	<0.0001 *	42.2 ± 5.41	44.1 ± 5.33	40.4 ± 4.88	0.001 *	38.8 ± 4.82	39.7 ± 4.79	38.2 ± 4.77	0.153
White blood cells, 10^3^/μL	8.36 ± 3.76	8.78 ± 4.50	8.01 ± 3.04	0.199	8.30 ± 3.25	8.51 ± 3.33	8.09 ± 3.21	0.569	8.41 ± 4.24	9.14 ± 5.70	7.95 ± 2.92	0.224
Platelets, 10^3^/μL	215 ± 90.79	207 ± 46.28	222 ± 114.44	0.280	211 ± 113.17	202 ± 42.69	220 ± 153.85	0.469	220 ± 60.21	215 ± 50.30	223 ± 66.11	0.531
Random glucose, mg/dL	148 ± 51.92	151 ± 51.07	145 ± 52.74	0.501	141 ± 40.61	143 ± 44.91	139 ± 36.39	0.650	154 ± 60.96	160 ± 57.41	150 ± 63.43	0.477
Total cholesterol, mg/dL	147 ± 39.79	155 ± 45.41	141 ± 33.28	0.029 *	141 ± 30.76	148 ± 34.74	133 ± 24.31	0.028 *	154 ± 46.56	163 ± 55.50	147 ± 38.51	0.161
LDL-cholesterol, mg/dL	86 ± 28.76	92 ± 30.78	81 ± 26.09	0.017 *	82 ± 22.99	88 ± 24.18	75 ± 20.07	0.013 *	90 ± 33.29	97 ± 37.36	86 ± 29.77	0.157
HDL-cholesterol, mg/dL	44 ± 10.76	45 ± 11.98	44 ± 9.70	0.844	43 ± 9.27	44 ± 9.97	42 ± 8.56	0.475	46 ± 11.97	46 ± 14.20	46 ± 10.33	0.917
Triglyceride, mg/dL	93 ± 48.19	93 ± 35.54	93 ± 56.84	0.967	90 ± 40.61	91 ± 34.30	89 ± 46.53	0.879	96 ± 55.00	95 ± 37.43	96 ± 64.81	0.939
INR	1.15 ± 0.27	1.05 ± 0.84	1.23 ± 0.33	<0.0001 *	1.16 ± 0.24	1.06 ± 0.08	1.25 ± 0.30	<0.0001 *	1.14 ± 0.29	1.04 ± 0.09	1.20 ± 0.35	0.002 *
SBV, cP	4.51 ± 0.57	4.74 ± 0.56	4.32 ± 0.51	<0.0001 *	4.64 ± 0.55	4.85 ± 0.57	4.44 ± 0.46	0.001 *	4.37 ± 0.56	4.61 ± 0.52	4.22 ± 0.53	0.002 *
DBV, cP	28.74 ± 8.28	31.92 ± 8.39	26.20 ± 7.30	<0.0001 *	30.79 ± 8.33	33.73 ± 8.88	27.93 ± 6.71	0.001 *	26.63 ± 7.73	29.59 ± 7.20	24.72 ± 7.52	0.005 *
hs-CRP, mg/dL	1.46 ± 2.67	1.92 ± 3.12	1.06 ± 0.21	0.058	1.74 ± 3.17	2.13 ± 3.42	1.35 ± 2.91	0.303	1.18 ± 2.03	1.68 ± 2.77	0.79 ± 1.05	0.096

Values are presented as mean ± standard deviation or numbers (%). LDL, low-density lipoprotein; HDL, high-density lipoprotein; INR, international normalized ratio; SBV, systolic blood viscosity; cP, centipoise; DBV, diastolic blood viscosity; hs-CRP, high sensitive C-reactive protein. * Significant *p*-value.

**Table 2 jcm-12-00887-t002:** Differences in blood viscosity according to the type of prior used antithrombotic agents.

	SBV, cP	*p*	DBV, cP	*p*
No antithrombotics (n = 71) vs.	4.74 ± 0.56		31.92 ± 8.39	
Prior antithrombotics (n = 89)	4.32 ± 0.51	<0.0001 *	26.20 ± 7.30	0.002 *
Antiplatelets (n = 51)	4.38 ± 0.54	0.003 *	26.92 ± 7.86	0.008 *
Aspirin (n = 35)	4.33 ± 0.52	<0.0001 *	26.09 ± 7.55	0.008 *
Clopidogrel (n = 11)	4.55 ± 0.48	0.275	29.69 ± 7.73	0.456
A + C (n = 5)	4.41 ± 0.78	0.014 *	26.65 ± 10.49	0.012 *
Warfarin (n = 11)	4.27 ± 0.63	0.012 *	26.65 ± 10.49	0.042 *
NOACs (n = 27)	4.22 ± 0.40	0.001 *	24.88 ± 5.33	0.002 *
Edoxaban (n = 17)	4.15 ± 0.41	0.005 *	23.99 ± 5.49	0.012 *
Rivaroxaban (n = 4)	4.13 ± 0.07	0.020 *	23.67 ± 0.57	0.034 *
Apixaban (n = 3)	4.17 ± 0.24	0.180	24.29 ± 2.77	0.230
Dabigatran (n = 3)	4.76 ± 0.46	0.143	32.09 ± 5.74	0.141
Antiplatelets vs. warfarin		0.720		0.842
Aspirin vs. warfarin		0.782		0.998
Clopidogrel vs. warfarin		0.340		0.372
A + C vs. warfarin		0.825		0.534
NOACs vs. antiplatelets		0.458		0.703
NOACs vs. warfarin		0.565		0.444

Values are presented as mean ± standard deviation. SBV, systolic blood viscosity; cP, centipoise; DBV, diastolic blood viscosity; A + C, aspirin plus clopidogrel; NOACs, new oral anticoagulants. * Significant *p*-value.

## Data Availability

The data that support the findings of this study are available from S.W.H., but restrictions apply to the availability of these data, which were used under license for the current study and so are not publicly available. Data are, however, available from the authors upon reasonable request and with the permission of S.W.H.

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
