# Peer review of "The Impact of Prior Antithrombotic Use on Blood Viscosity in Cardioembolic Stroke with Non-Valvular Atrial Fibrillation"

_jcm, 2023, doi:10.3390/jcm12030887_

Round 1
Reviewer 1 Report
This paper reports on the results a single center retrospective study of effects taking anti-thrombotic on systolic and diastolic blood viscosity (BV) among patients presently with presumed atrial fibrillation-associated acute ischaemic stroke. The authors found that antithrombotic use was associated with reduced BV
There are a number of issue the authors may wish to address
1. 1st author’s affiliation - line 5 – city and country left out
2. Abstract – if the focus is on NOACs, this is buried under the lumping of the data with the other antithrombotics, and thus does not support the conclusion stated in lines 30,31. The authors need to be clear what they are studying and show this information clearly
3. Introduction – the need to separately study the effects on SBV and DBV is not explained
4. Methods – lines 74,75 - how did the authors decide that the stroke was ‘presumably due to an embolus arising from NVAF’? lines 89-91 - how was compliance with medication prior to the stroke ascertained? Records of medication prescription only is not be enough. Lines 98,99 – how did the authors decide that 300 s-1 and 1 s-1 adequately represented SBV and DBV? What is ‘normal’ SBV DBV?
5. Results – SBV EBV lines 130,131 are inconsistent with lines 135,136, 160. Line 151 – what are the ‘these groups’ the authors are referring to?
6. Discussion - the 1st para should emphasise the results pertinent to the main aim of the study ie. the effects of NOACs. There is no discussion of the impact on SBV vs DBV, and its clinical relevance. Another limitation is also that the number of patients taking each type of antithrombotic is small, thus comparison is difficult eg apixaban, dabigatran. The clinical value of this study would be by having stroke-free controls, or showing that those with lower SBV and DBV have less stroke or less severe stroke – else this research is just showing lab data of no clinical relevance, as the patients still had stroke…..
7. Conclusions – lines 223,224 – the authors did not study this….lines 225,226 and Abstract line 31 – I do not know if reduced BV is an ‘improve’ment….
Author Response
This paper reports on the results a single center retrospective study of effects taking anti-thrombotic on systolic and diastolic blood viscosity (BV) among patients presently with presumed atrial fibrillation-associated acute ischaemic stroke. The authors found that antithrombotic use was associated with reduced BV.
- 1st author’s affiliation - line 5 – city and country left out
Response: Thanks for reviewer’s detailed review of our manuscript. According to reviewer’s comment, we filled out the omitted information in the author’s affiliations.
Department of Neurology, Gangnam Severance Hospital, Yonsei University College of Medicine, Seoul, Korea, yhjung@yuhs.ac
- Abstract – if the focus is on NOACs, this is buried under the lumping of the data with the other antithrombotics, and thus does not support the conclusion stated in lines 30,31. The authors need to be clear what they are studying and show this information clearly
Response: Reviewer’s suggestion is a very insightful point. At the beginning of the study, we have attempt to determine the effects of previous NOACs use on BV in patients with acute CES due to NVAF. However, we should broaden the study population from NOAC patients to all antithrombotic patient as the small number of enrolled patient who had taken NOACs. Thanks to reviewer’s valuable comments, we revised the title and clarified the aim of this study in abstract and introduction, and discussed about these issues raised by reviewer, and modified our conclusion in revised manuscript as following :
Title
The Impact of Prior Antithrombotic use on Blood Viscosity in Cardioembolic Stroke with Non-Valvular Atrial Fibrillation
Abstract
Abstract: Although clinical studies have demonstrated prior use of antiplatelets was associated with decreased BV in patients with acute ischemic stroke, the impact of previous anticoagulants use on blood viscosity in cardioembolic stroke with non-valvular AF (NVAF) has not yet been clearly studied. This single-center retrospective observational study aimed to determine the impact of prior antithrombotic (antiplatelet and anticoagulant) use on BV in patients with cardioembolic stroke (CES) due to NVAF. Patients with CES and NVAF were analyzed with the following inclusion criteria: 1) patients over 20 years of age admitted within five days of stroke onset; 2) ischemic stroke presumably due to an NVAF-derived embolus; 3) compatible cortical/subcortical lesion on brain computed tomography or magnetic resonance imaging; 4) hemoglobin level of 10-18 mg/dL; and 5) receiving antiplatelets within five days or anticoagulants within two days if previously medicated. From the screening of 195 patients (22% of the total stroke population during the study period), who had experienced ischemic stroke with AF, 160 were included for the final analysis. Eighty-nine patients (56%) were taking antithrombotics (antiplatelet, 57%; warfarin, 13%; NOACs, 30%) regularly. Compared to patients without previous antithrombotic use, those with previous antithrombotic use (antiplatelets, warfarin, and NOACs) were significantly associated with decreased systolic BV (SBV) and diastolic BV (DBV) (P<0.036). In multiple linear regression analysis, hematocrit (Hct) level and prior antithrombotic use was significantly associated with decreased SBV and DBV. Hct was positively correlated with increased SBV and DBV. In Hct-adjusted partial correlation analysis, prior uses of any antithrombotic agents were associated with decreased SBV (r<-0.270, P<0.015) and DBV (r<-0.183, P<0.044). In conclusion, this study showed that prior antithrombotic use (antiplatelets, VKAs, and NOACs) was associated with decreased SBV and DBV in patients presenting with acute CES secondary to NVAF. Our results indicated that previous use of NOACs may be a useful hemorheological parameter in patients with acute CES due to NVAF. Accumulation of clinical data from a large number of patients with the risk of stroke occurrence, initial stroke severity, and functional outcome is necessary to assess the usefulness of BV.
Introduction
Blood viscosity (BV), the key predictor of endothelial shear stress, is defined as intrinsic resistance applied to blood flow [7]. BV determines the frictional force applied to the blood vessel wall and is characterized by blood stickiness and thickness [7, 8]. As an important hemorheological parameter, BV is determined by hematocrit (Hct) level, plasma viscosity (PV), and erythrocyte deformability [7]. Diastolic BV (DBV), which represents the hemorheological state of blood corresponding to a very slow flow, becomes a critical factor for determining the tissue perfusion status and could more considerably impact microvascular perfusion than SBV in acute ischemic stroke [9]. Elevated BV, which involves increased Hct and PV, and decreased erythrocyte deformability, increases thromboembolic risk. Therefore, elevated BV is related to an increased risk of cardiovascular events and may contribute to ischemic stroke occurrence, in addition to all-cause mortality [9, 10].
In AF patients, several studies suggested that hyperviscosity due to hemorheological alterations was frequent [9, 11, 12]. One study showed that increased BV at high shear rates and reduced erythrocyte deformability were independently related to ischemic stroke occurrence in AF patients [9]. Regarding treatment, VKAs reduce BV significantly in patients with acute ischemic stroke and NVAF [11]. We previously demonstrated that prior use of antiplatelets was associated with decreased BV in patients with acute ischemic stroke [13]. However, the impact of prior antithrombotic use on BV in patient with NVAF has not yet been elucidated. Therefore, this study aimed to determine the effects of prior antithrombotic use on BV in patients with acute CES due to NVAF
Disccusion
This study investigated the impact of prior antithrombotic use on BV in patients presenting with acute CES secondary to NVAF. Interestingly, compared to the group without prior antithrombotic use, all groups with prior antithrombotic use (antiplatelets, VKAs, and NOACs) were significantly associated with decreased SBV and DBV. There was no difference in SBV and DBV according to the type of antithrombotic agents used.
Conclusion
This study showed that prior antithrombotic use (antiplatelets, VKAs, and NOACs) was associated with decreased SBV and DBV in patients presenting with acute CES secondary to NVAF. AF alters the composition of blood clotting factors, such as fibrin and thrombin-antithrombin complexes, which promote thrombus formation. Our results indicated that previous use of NOACs may be a useful hemorheological parameter in patients with acute CES due to NVAF. Accumulation of clinical data from a large number of patients with the risk of stroke occurrence, initial stroke severity, and functional outcome is necessary to assess the usefulness of BV.
- Introduction – the need to separately study the effects on SBV and DBV is not explained
Response: According to reviewer’s critical comment, we wanted to find and explain the effects on SBV and DBV in this study, but we could not find a relevant paper. To our knowledge, the reason for dividing into SBV and DBV on research of blood viscosity in patients with stroke is limited and not clearly known. However, most of the blood viscosity reported that DBV changes are related to microvessel perfusion, that is lacunar stroke. This was added to the introduction description of the BV. Description of BV (SBV and DBV) measurement was performed in BV measurement of Method (line 94~101). The sharp point of the reviewers makes the meaning clear.
Diastolic BV(DBV), which represents the hemorheological state of blood corresponding to a very slow flow, becomes a critical factor for determining the tissue perfusion status and could more considerably impact microvascular perfusion than SBV in acute ischemic stroke.
- Methods – lines 74,75 - how did the authors decide that the stroke was ‘presumably due to an embolus arising from NVAF’? lines 89-91 - how was compliance with medication prior to the stroke ascertained? Records of medication prescription only is not be enough. Lines 98,99 – how did the authors decide that 300 s-1 and 1 s-1 adequately represented SBV and DBV? What is ‘normal’ SBV DBV?
Response: Thanks for reviewer’s detailed review of our data and valuable comments.
4-1 how did the authors decide that the stroke was ‘presumably due to an embolus arising from NVAF’?
As reviewer point out, just because a patient with cerebral infarction with NVAF does not mean that the etiology of stroke is due to embolism caused by NVAF. The existing inclusion criteria were further modified to include patients according to the cardioembolic stroke criteria presented in the TOAST classification. We clarified the stroke etiology subgroup as following; “2) ischemic stroke presumably due to an embolus arising from NVAF stroke according to the Trial of ORG 10172 in the Acute Stroke Treatment Classification.
4-2 how was compliance with medication prior to the stroke ascertained? Records of medication prescription only is not be enough.
Although the measurement of patients' adherence to treatment is difficult, checking the adherence of medication is an important point of this analysis. Unfortunately, this study is not a questionnaire survey to confirms adherence to treatment. To check medication adherence, we added that the physician checked the recent drug use history obtained through intimate interviews with patients or caregivers when visiting the hospital. It was routine practice in our study hospital for physician or nurses to inquire about the prior medication status at the time of admission with their next of kin or caregiver as well as with the patients themselves
Method
Medications taken regularly during the week preceding admission were recorded by a physician who interview with patients or caregivers when visiting the hospital and by a skilled pharmacist based on the hospital records.
4-3 how did the authors decide that 300 s-1 and 1 s-1 adequately represented SBV and DBV?
In this study, whole BV (WBV) was measured using a computerized scanning capillary-tube viscometer (SCTV) (Hemovister, Pharmode Inc., Seoul, Korea). BV measured at shear rates of 300 s−1 and 1 s−1 were adequately presented SBV and DBV in previous studies using this machine. (Rosuvastatin Reduces Blood Viscosity in Patients with Acute Coronary Syndrome; Korean Circulation Journal 2016; 46(2): 147-153, Analytical performance evaluation of the scanning capillary tube viscometer for measurement of whole blood viscosity; Clinical biochemistry 2013; 46(1-2): 139-142). In order to evaluate SBV and DBV in this study, we used the existing research method. The relevant content in the referenced paper is extracted and presented to the reviewer.
“In the present study, the WBV measured at a shear rate of 300 s− 1 is referred as the systolic whole blood viscosity (SBV), whereas that measured at a shear rate of 1 s− 1 is referred as the diastolic whole blood viscosity (DBV).”
“The SCTV had good within-run and total-run coefficient of variant (CV)s at low-, medium-, and high-concentration samples, at shear rates of 1 and 300 s− 1. The within-day CVs with the three human blood samples were 6.3%, 3.7% and 3.8% at a shear rate of 1 s− 1, and 3.2%, 3.0% and 4.1% at a shear rate of 300 s− 1. The SCTV method showed an excellent linearity in the range of 84.9 to 558.2 milliPoise (mP) and 28.8 to 71.0 mP at shear rates of 1 and 300 s− 1, respectively.”
We have revised method regarding for reference as following;
- Song SH, Kim JH, Lee JH, Yun YM, Choi DH, Kim HY. Elevated blood viscosity is associated with cerebral small vessel disease in patients with acute ischemic stroke. BMC Neurol. 2017;17:20.
à 16. Kim HN, Cho YI, Lee DH, Park CM, Moon HW, Hur MN, et al. Analytical performance evaluation of the scanning capillary tube viscometer for measurement of whole blood viscosity. Clinical biochemistry. 2013;46:139-142
4-4 What is ‘normal’ SBV DBV?
Although normal value of BV were reported in research of cardiovascular or other disease, in current clinical practice, several methods (including our SCTV) for the assessment of WBV exist and every pathology unit has equipment for any of the diverse methods. Therefore, it isn't easy to present normal values of SBV and DBV in this research method.
- Results – SBV EBV lines 130,131 are inconsistent with lines 135,136, 160. Line 151 – what are the ‘these groups’ the authors are referring to?
Response: We checked the contents mentioned by the reviewer again. Throughout this study, this research consistently confirmed that the BV is lower in the group taking antithrombotic drugs before the occurrence of CES (line 130.131). And We interpreted that this trend (BV is lower in the group taking antithrombotic drug) appears in the analysis of men and women as well (line 135.136). (Table 1)
On the other hand, when the BV difference according to the type of antithrombotic agents (taking antiplatelets and anticoagulant) was analyzed in detail, there was no BV difference between the antiplatelet, VKAs and NOACS group. (Table 3) As the reviewer pointed out, the expression “these group” can lead to confusing conclusions for the reader. We clarified the results about Table 3 in revised manuscripts as following (line 151): ‘There were no differences in SBV and DBV among antiplatelet, VKAs and NOACs these groups.’
- Discussion - the 1st para should emphasise the results pertinent to the main aim of the study ie. the effects of NOACs. There is no discussion of the impact on SBV vs DBV, and its clinical relevance. Another limitation is also that the number of patients taking each type of antithrombotic is small, thus comparison is difficult eg apixaban, dabigatran. The clinical value of this study would be by having stroke-free controls, or showing that those with lower SBV and DBV have less stroke or less severe stroke – else this research is just showing lab data of no clinical relevance, as the patients still had stroke…..
Response: We thank the reviewer for advice and comments.
According to reviewer’s comment, we revised the first paragraph in the disccusion as following;
This study investigated the impact of prior antithrombotic use on BV in patients presenting with acute CES secondary to NVAF. Interestingly, compared to the group without prior antithrombotic use, all groups with prior antithrombotic use (antiplatelets, VKAs, and NOACs) were significantly associated with decreased SBV and DBV. There was no difference in SBV and DBV according to the type of antithrombotic agents used.
In agreement with the reviewer’s comment, the number of patients taking each type of antithrombotic, especially NOACs, is small. Thus, the comparison is difficult. This reason makes us change the purpose of this study, "the impact of NOAC,” to "the impact of antithrombotics” on blood viscosity. We added this point as one of limitations in this study as following;
Our study had several limitations. First, the number of patients included the prior NOAC group (edoxaban n=17, rivaroxaban n=4, apixaban n=3, and dabigatran n=3) was not large in this retrospective study, which could implying the existence of unmeasured biases and affect the validity of the results. We also could not obtain important clinical data, such as initial stroke severity and outcome. Therefore, we cannot demonstrate whether those SBV and DBV in patients with CES differences have clinical implications.
- Conclusions – lines 223,224 – the authors did not study this….lines 225,226 and Abstract line 31 – I do not know if reduced BV is an ‘improve’ment….
Response: Thank you for reviewer’s comments. We understand the reviewer’s point, deleted the unnecessary sentence, and modified our conclusion in the revised manuscript as following :
This study showed that prior antithrombotic use (antiplatelets, VKAs, and NOACs) was associated with decreased SBV and DBV in patients presenting with acute CES secondary to NVAF. AF alters the composition of blood clotting factors, such as fibrin and thrombin-antithrombin complexes, which promote thrombus formation. Our results indicated that previous use of NOACs may be a useful hemorheological parameter in patients with acute CES due to NVAF. Accumulation of clinical data from a large number of patients with the risk of stroke occurrence, initial stroke severity, and functional outcome is necessary to assess the usefulness of BV.

Reviewer 2 Report
The authors performed a retrospective biobank analysis of patients being admitted to the hospital with acute stroke. They found that blood viscosity was significantly lower in patients taking antiplatelet medication, including new oral anticoagulants.
The manuscript is well written with adequate statistics and references. However, in my opinion there are too many tables (tables 2 and 3 may be moved to the appendix).
Another major concern is that title and abstract both focus on NOACs only, while in the rest of the manuscript patients with vs. without any antiplatelet agent. The authors should therefore either change the title and abstract, or adapt the analysis accordingly.
It is not clear to me how the blood was drawn without the consent of the patient – was there some kind of biobank involved? Did the patients at least consent drawing of blood?
Author Response
Reviewer 2
- The authors performed a retrospective biobank analysis of patients being admitted to the hospital with acute stroke. They found that blood viscosity was significantly lower in patients taking antiplatelet medication, including new oral anticoagulants. The manuscript is well written with adequate statistics and references. However, in my opinion there are too many tables (tables 2 and 3 may be moved to the appendix).
Response: To comply with the reviewer’s comment, we moved table 2 and 3 to the appendix. This appears in the revised manuscript. (page 7, line 279-81)
- Another major concern is that title and abstract both focus on NOACs only, while in the rest of the manuscript patients with vs. without any antiplatelet agent. The authors should therefore either change the title and abstract, or adapt the analysis accordingly.
Response: Thanks for reviewer’s detailed review of our data and valuable comments. There was also a comment from the other reviewer, and we understand and agree with the comments of the two reviewers. At first, we have attempt to determine the effects of previous NOACs use on BV in patients with acute CES due to NVAF. However we should broden the study population from NOAC patient to all antithrobotics patient as small number of inrolled patient who had taken NOACs. Thanks to reviewer’s valuable comments, we revised the title and clarified the aim of this study in abstract and introduction, and discussed about these issues raised by reviewer, and modified our conclusion in revised manuscript as following :
Title
The Impact of Prior Antithrombotic use on Blood Viscosity in Cardioembolic Stroke with Non-Valvular Atrial Fibrillation
Abstract
Abstract: Although clinical studies have demonstrated prior use of antiplatelets was associated with decreased BV in patients with acute ischemic stroke, the impact of previous anticoagulants use on blood viscosity in cardioembolic stroke with non-valvular AF (NVAF) has not yet been clearly studied. This single-center retrospective observational study aimed to determine the impact of prior antithrombotic (antiplatelet and anticoagulant) use on BV in patients with cardioembolic stroke (CES) due to NVAF. Patients with CES and NVAF were analyzed with the following inclusion criteria: 1) patients over 20 years of age admitted within five days of stroke onset; 2) ischemic stroke presumably due to an NVAF-derived embolus; 3) compatible cortical/subcortical lesion on brain computed tomography or magnetic resonance imaging; 4) hemoglobin level of 10-18 mg/dL; and 5) receiving antiplatelets within five days or anticoagulants within two days if previously medicated. From the screening of 195 patients (22% of the total stroke population during the study period), who had experienced ischemic stroke with AF, 160 were included for the final analysis. Eighty-nine patients (56%) were taking antithrombotics (antiplatelet, 57%; warfarin, 13%; NOACs, 30%) regularly. Compared to patients without previous antithrombotic use, those with previous antithrombotic use (antiplatelets, warfarin, and NOACs) were significantly associated with decreased systolic BV (SBV) and diastolic BV (DBV) (P<0.036). In multiple linear regression analysis, hematocrit (Hct) level and prior antithrombotic use was significantly associated with decreased SBV and DBV. Hct was positively correlated with increased SBV and DBV. In Hct-adjusted partial correlation analysis, prior uses of any antithrombotic agents were associated with decreased SBV (r<-0.270, P<0.015) and DBV (r<-0.183, P<0.044). In conclusion, this study showed that prior antithrombotic use (antiplatelets, VKAs, and NOACs) was associated with decreased SBV and DBV in patients presenting with acute CES secondary to NVAF. Our results indicated that previous use of NOACs may be a useful hemorheological parameter in patients with acute CES due to NVAF. Accumulation of clinical data from a large number of patients with the risk of stroke occurrence, initial stroke severity, and functional outcome is necessary to assess the usefulness of BV.
Introduction
Blood viscosity (BV), the key predictor of endothelial shear stress, is defined as intrinsic resistance applied to blood flow [7]. BV determines the frictional force applied to the blood vessel wall and is characterized by blood stickiness and thickness [7, 8]. As an important hemorheological parameter, BV is determined by hematocrit (Hct) level, plasma viscosity (PV), and erythrocyte deformability [7]. Diastolic BV (DBV), which represents the hemorheological state of blood corresponding to a very slow flow, becomes a critical factor for determining the tissue perfusion status and could more considerably impact microvascular perfusion than SBV in acute ischemic stroke [9]. Elevated BV, which involves increased Hct and PV, and decreased erythrocyte deformability, increases thromboembolic risk. Therefore, elevated BV is related to an increased risk of cardiovascular events and may contribute to ischemic stroke occurrence, in addition to all-cause mortality [9, 10].
In AF patients, several studies suggested that hyperviscosity due to hemorheological alterations was frequent [9, 11, 12]. One study showed that increased BV at high shear rates and reduced erythrocyte deformability were independently related to ischemic stroke occurrence in AF patients [9]. Regarding treatment, VKAs reduce BV significantly in patients with acute ischemic stroke and NVAF [11]. We previously demonstrated that prior use of antiplatelets was associated with decreased BV in patients with acute ischemic stroke [13]. However, the impact of prior antithrombotic use on BV in patient with NVAF has not yet been elucidated. Therefore, this study aimed to determine the effects of prior antithrombotic use on BV in patients with acute CES due to NVAF
Disccusion
This study investigated the impact of prior antithrombotic use on BV in patients presenting with acute CES secondary to NVAF. Interestingly, compared to the group without prior antithrombotic use, all groups with prior antithrombotic use (antiplatelets, VKAs, and NOACs) were significantly associated with decreased SBV and DBV. There was no difference in SBV and DBV according to the type of antithrombotic agents used.
Conclusion
This study showed that prior antithrombotic use (antiplatelets, VKAs, and NOACs) was associated with decreased SBV and DBV in patients presenting with acute CES secondary to NVAF. AF alters the composition of blood clotting factors, such as fibrin and thrombin-antithrombin complexes, which promote thrombus formation. Our results indicated that previous use of NOACs may be a useful hemorheological parameter in patients with acute CES due to NVAF. Accumulation of clinical data from a large number of patients with the risk of stroke occurrence, initial stroke severity, and functional outcome is necessary to assess the usefulness of BV.
- It is not clear to me how the blood was drawn without the consent of the patient – was there some kind of biobank involved? Did the patients at least consent drawing of blood?
Response: We thank the reviewer for advice. This research was a retrospective medical record study. In addition, we thought that the risk to the study subjects was low even if consent was waived and the data were studied by thoroughly anonymizing the patient's personal information. As pointed out, the sentence of the omission from informed consent were revised as follows;
The informed consent requirement was waived because the database was accessed only for analysis. This study removed individual identifying variables, and the researcher could secure only the results.

Round 2
Reviewer 1 Report
The authors have adequately responded to my concerns
Author Response
Thanks for reviewer’s detailed review of our data and valuable comments.